# Improvement of Torque Estimation for Series Viscoelastic Actuator Based on Dual Extended Kalman Filter

**Hui Wei, Kui Xiang, Haibo Chen, Biwei Tang and Muye Pang ***

School of Automation, Wuhan University of Technology, Wuhan 430070, China; weihui@whut.edu.cn (H.W.); xkarcher@126.com (K.X.); chen_hai_bo@whut.edu.cn (H.C.); tangbiwei@whut.edu.cn (B.T.)
* Correspondence: pangmuye@whut.edu.cn

**Abstract:** Adding damping such as viscoelastic element in series elastic actuators (SEA) can improve the force control bandwidth of the system and suppression of high frequency oscillations induced by the environment. Thanks to such advantages, series viscoelastic actuators (SVA) have recently gained increasing research interests from the community of robotic device design. Due to the inconvenience of mounting torque sensors, employing the viscoelastic elements to directly estimate the output torque is of great significance regarding the real-world applications of SVA. However, the nonlinearity and time-varying properties of viscoelastic materials would degrade the torque estimation accuracy. In such a case, it is paramount to simultaneously estimate the output torque state and viscoelastic model coefficients in order to enhance the torque estimation accuracy. To this end, this paper first completed the design of a rubber-based SVA device and used the Zenner linear viscoelastic model to model the viscoelastic element of the rubber. Subsequently, this paper proposed a dual extended Kalman filter- (DEFK) based torque estimation method to estimate the output torque and viscoelastic model coefficients simultaneously. The noisy observations of two Kalman filters were provided by motor current-based estimated torque. Moreover, the dynamic friction of harmonic drive of the designed SVA was modeled and compensated to enhance the reliability of current-based torque estimation. Finally, a number of experiments were carried out on SVA, and the experimental results confirmed the DEFK effectiveness of improving torque estimation accuracy compared to only-used rubber and only-used motor current torque estimation methods. Thus, the proposed method could be considered as an effective alternative approach of torque estimation for SVA.

**Keywords:** SVA; output torque estimation; dual extended Kalman filter; rubber model; harmonic drive friction compensation

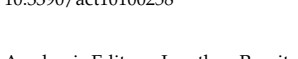

## 1. Introduction

Thanks to its relative low output impedance and convenience of torque measurement, series elastic actuators (SEA) are commonly used in the field of robot joint force control in recent years [1–3]. The metal springs used in SEA exhibit linear relationships between output force and position, which greatly increases torque control accuracy owing to the ability of turning the force control problem into a position control problem [4]. Nevertheless, the remaining small inertias caused by the elastic element and the end effector of SEA can result in undesired vibrational effects in the case where SEA works with high operation frequencies under dynamic robot-environment interaction circumstances [5]. Therefore, there exists strong necessity to address the issue of vibration suppression with respect to SEA [6–9].

Adding damping characteristics in SEA, which makes a new form of actuator called series viscoelastic actuator (SVA), is one of the efficient approaches to decrease high-frequency oscillations arising from impacts [10,11]. Due to this advantage of SVA, it has aroused increasing research interests in recent years. For example, Kim et al. proved that adding damping to SEA can effectively reduce the relative degree of torque dynamics

order by one, thereby improving the anti-interference ability of the actuator under hard impacts [12]. Nonetheless, the device developed by Kim would be not simple, or compact enough to be embedded in robot joints.

Rubber could be a vital alternative to solve the issue of compact design of SEA since this material not only has the elastic and damping properties, but also is tiny in its size [13]. Besides, implementation of rubber to compose SVA can notably improve the force control bandwidth and suppress the undesired vibrational effects [14]. In virtue of these advantages of rubber, it has been applied to design SVA in recent years [15]. However, nonlinearity and time-varying properties of the rubber impose great challenges on torque estimation of SVA [16].

In order to improve the torque estimation accuracy of SVA, it is necessary to model the nonlinearity and time-varying properties of rubber material. Commonly used models of viscoelastic materials include Maxwell model, Kelvin–Voigt model, Zenner model and Burgers model, etc. These models are composed by a series or parallel of spring and dashpot elements to describe various behaviors of viscoelastic materials. Kelvin–Voigt model can describe the force creep, but not the stress relaxation, of the viscoelastic materials, while Maxwell model focuses on stress relaxation but not force creep. Zenner model and Burgers model include the above two characteristics [17], but Burgers model has higher-order derivatives on force and deflection, resulting in weak resistance to amplified noise in real-world implementations [14]. Taking into account the integrity of various rubber characteristics and the stability for applications, Zenner model can be used as a logical model to estimate the output torque of the rubber element in the SVA. However, the intrinsic properties of rubber material are strongly affected by environment factors such as temperature, which means the parameters of the model are time variant. How to capture the time-variant effect of rubber becomes the key point for improving torque estimation accuracy.

Motor current could also be used to estimate motor torque, however, the torque provided by the motor alone is too small to fulfill the application requirement. The harmonic drive is often adopted to play the role in amplifying the motor's output torque [18,19]. As a consequence, it is essential to model the nonlinearities of harmonic drive in order to obtain more accurate torque estimation of SVAs. The mechanism of friction in the harmonic drive is extensively investigated in recent years [20–22]. Marilier et al. showed the relationship between the friction torque of the harmonic drive and the motor speed [23]. Gandhi et al. revealed that the friction torque of harmonic drive was not only nonlinearly related to motor speed, but also periodically related to motor position [24]. Donghui represented that the friction torque had Stribeck effect at low speed [25]. It is notable that the performance of the current-based torque estimation method for SVA composed by harmonic reducer could be unpromising in the case where the acceleration of motor changes drastically, even though the nonlinearities are well modeled and compensated.

In many applications, it is convenient and desirable to estimate the output torque from viscoelastic element because commercial torque sensors are fragile to impact and expensive. The contribution of this paper is to compensate nonlinearity of harmonic driver and time-variant properties of rubber by a dual extended Kalman filter to improve output torque estimation accuracy of SVA in real time. The idea of DEKF [26,27] is derived from a phenomenon that rubber-based torque estimation is reliable in a short period but inaccurate after working for a long period of time, while current-based torque estimation is more accurate when motor works at steady state but incorrect when motor works at dynamic situation. The rubber is modeled by Zenner linear viscoelastic model while the friction of harmonic drive is modeled based on harmonic drive output shaft position and velocity-related functions. One extended Kalman filter works as a state estimator to compensate the nonlinear error of harmonic drive for non-constant acceleration conditions of the motor, and the other one works as a parameter estimator to correct the parameters of the rubber in real time to ensure the accuracy of torque estimation from rubber-based method. Two EKFs take the rubber-based estimated torque and the parameters of the rubber model as

SVA system states, respectively. Observation values for the two filters are provided only by current-based torque estimation. A quantitative confidence function $\psi$, which is referred from [28], with an interval range of [0, 1] is defined to judge the accuracy of the current-based estimated torque. When $\psi$ tends to 0 which means current-based estimated torque is unreliable, the update status of two filters are mainly determined by the rubber-based torque estimation model. When $\psi$ tends to 1, the update status are determined by fusion of rubber-based and current-based values. The experimental results reveal that the torque after DEKF compensation has a more accurate torque output capability (0.027 Nm root mean square error and 1.48%/FS nonlinear error), compared to only using rubber (0.029 Nm root mean square error and 2.38%/FS nonlinear error) and current (0.178 Nm root mean square error and 18.6%/FS nonlinear error).

## 2. Materials and Methods

### 2.1. SVA's Mechanical Structure

The self-designed SVA is composed by mainly three parts, a BLDC motor (MAXON, EC45), a harmonic drive (Leaderdrive, LHS-I-14) and a rubber element (as shown in Figure 1). The motor and harmonic drive are connected together to form a torque generator which is deployed on the left side of SVA. Rubber material (natural rubber with shore hardness of 50 A, provided by Northwest Rubber & Plastics Research & Design Institute, Xi'an, China) is utilized as the viscoelastic part deployed on the right side with the form of a cylinder. The size of the cylinder is 30 mm in diameter and 10 mm in thickness and the shore hardness of rubber is 50 A. The lower surface of the rubber adheres to the output shaft of SVA, and the upper surface adheres to the transmission gear surface at the output end of the harmonic drive. The upper surface of rubber is defined as the input and the lower surface is defined as the output. Two quadrature photoelectric encoders are placed on both upper and lower sides of the rubber. The rotation angle is collected by the single-chip microcomputer, and the difference between the two angular positions is taken as the rubber deformation.

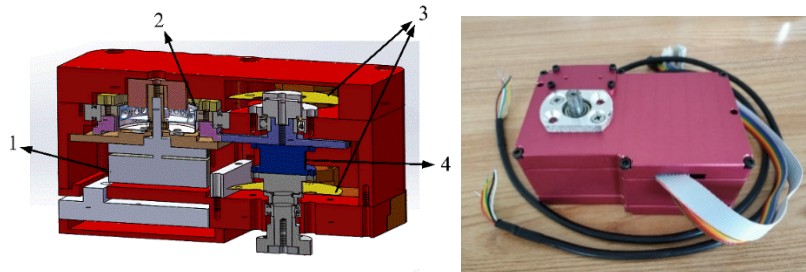

**Figure 1.** Mechanical structure of our self-designed SVA. The 3D mechanical model of SVA is shown on the left of this figure: (1) motor, (2) harmonic drive, (3) encoder, (4) rubber.

### 2.2. Dynamic Model of SVA

The simplified schematic diagram of the dynamic model of the SVA device is shown in Figure 2. Due to its good manufacturing process and smaller electrical time constant than the mechanical time constant, the motor can be simplified as a torque source with certain inertia. Although the harmonic drive has a certain degree of flexibility, it is supposed as a rigid body which only contains damping and inertial characteristics because the rubber has higher order of magnitude compliant than harmonic drive. The rubber model adopts the Zenner model. The damping of rubber is caused by the internal friction of rubber molecules belonging to internal damping [29] and it does not cause system torque loss. Therefore, the output torque of the harmonic drive is approximately equal to the load torque. Thus, the dynamic model of SVA can be formulated as follows:

$$NT_m - T_l = (NJ_0 + J_1)\ddot{\varphi} + D(\dot{\varphi}, \varphi)\dot{\varphi} \tag{1}$$

where $T_m$ and $T_l$ are the motor torque and load torque respectively, $J_0$ is the inertia of the motor rotor, damping $D$ includes the damping of the harmonic drive and other rotating pairs of the system which is usually a nonlinear function of $\dot{\varphi}$. $J_1$ includes the inertia of harmonic drive. $N$ is the reduction ratio. $\ddot{\varphi}, \dot{\varphi}, \varphi$ are angular acceleration, angular velocity and angular position, respectively.

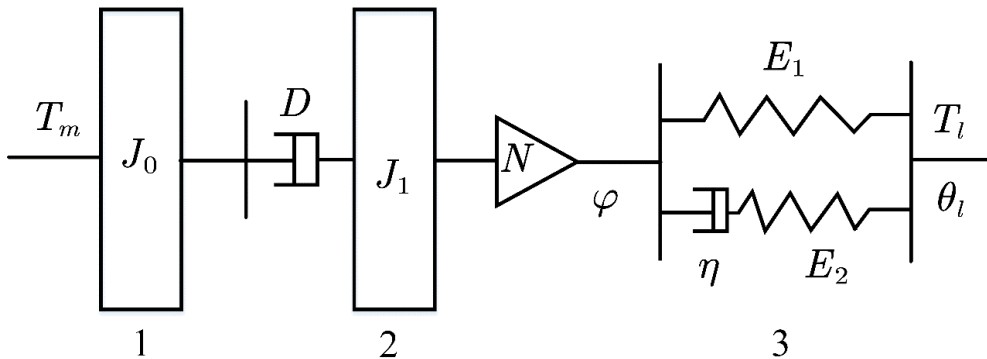

**Figure 2.** The simplified schematic diagram of the dynamic model of the SVA: (1) motor, (2) harmonic drive, (3) rubber. $T_m$ and $T_l$ are the motor torque and load torque respectively, $J_0$ includes the inertia of the motor rotor and harmonic drive, damping $D$ includes the damping of the harmonic drive and other rotating pairs of the system which is usually a nonlinear function of $\dot{\varphi}$. $J_1$ includes the inertia of harmonic drive. $N$ is the reduction ratio. $E_1, E_2, \eta$ are the rubber model parameters. $\ddot{\varphi}, \dot{\varphi}, \varphi$ are angular acceleration, angular velocity and angular position, respectively.

The motor output torque $T_m$ is proportional to the q-axis torque current $I_q$ which is calculated through Clarke and Park transform by the three-phase current collected from the motor drive board. The proportional coefficient is the motor torque constant $C$. The first item on the right side of Equation (1) is the torque loss caused by the inertias of the motor and the harmonic drive for acceleration. The second term on the right is the torque loss caused by damping, which originates from the harmonic drive and other motion pairs. The influence of the damping term is mainly manifested as the system friction torque $T_f$. It can be indicated from Equation (1) that the torque estimation accuracy depends on the measurement error of the inertia $J_0, J_1$, angular velocity acceleration $\ddot{\varphi}$, and damping coefficient. The inertias of motor and harmonic drive are difficult to measure accurately. The angular accelerations which are obtained by second-order differential of angular values are unable to be used because the measurement noises from encoder will be amplified by two orders. As a consequence, the estimation error of the inertia term $(NJ_0 + J_1)\ddot{\varphi}$ is usually too large to be accepted. The reliable way to acquire output torque of SVA by motor current is to perform the calculation when angular acceleration equals to zero. In such a situation, the dynamic model of SVA can be simplified as Equation (2) in which the inertial term is out of consideration.

$$T_l = NCI_q - D(\dot{\varphi}, \varphi)\dot{\varphi} \tag{2}$$

### 2.3. Model of the Rubber Element

An alternative way to directly acquire the output torque $T_l$ of SVA is to utilize the rubber element. The rubber-based torque estimation is based on the constitutive equations [30]. Parameters of the constitutive equation are derived from different geometric structures with the help of tools such as calculus to show the macroscopic mechanical properties of the rubber.

The left picture of Figure 3 shows a model figure of the designed rubber viscoelastic element which has a simple geometric shape, a cylinder with a radius of $R$ and a height of $H$. The rubber used in SVA is shown in the right of Figure 3. Input torque $T$ produces torsional deformation $\delta$ on the rubber cylinder part. A micro ring with a radius of $r$ and a width of $dr$ is taken on the torsion end surface. It can be assumed that the stress $\sigma$ and the shear strain $\varepsilon$ of each point on the micro ring are identical because of the occurrence

of concentric torsion. During the deforming process, the shear strain $\varepsilon$ can be obtained at point $B$ according to the geometric relationship of the rubber cylinder. Since the micro ring width $dr$ is very small, the area $S$ can be approximated to be $2\pi r dr$. Assuming that the shear stress at each point of the micro-ring is $\sigma$ and the arm length is the radius $r$, micro torque and shear strain can be calculated as follows:

$$\begin{cases} S = 2\pi r dr \\ \tau = \sigma S r = 2\pi r^2 \sigma dr \\ \varepsilon = \frac{\overset{\frown}{AB}}{AC} = \frac{r\delta}{H} \end{cases} \tag{3}$$

where $AC$, $\delta$, $r$, $\sigma$ are arc length, thickness of rubber, torsional deformation, radius, and stress, respectively. The rubber material satisfies the Zenner model, and its constitutive equation is shown as follows:

$$\frac{\eta\dot\sigma}{E_2} + \sigma = \frac{E_1 + E_2}{E_2}\eta\dot\varepsilon + E_2\varepsilon \tag{4}$$

where $\eta$, $\sigma$, $\varepsilon$ are damping coefficient, stress, strain of Zenner model, respectively, and $E_1$ and $E_2$ are elastic coefficient of Zenner model. Substituting Equation (3) into Equation (4), constitutive equation of the Zenner model can be transformed as follows:

$$\frac{\eta}{E_2}\dot\tau + \tau = (\frac{E_1 + E_2}{E_2}\eta\dot\delta + E_2\delta)\frac{2\pi r^3}{H}dr \tag{5}$$

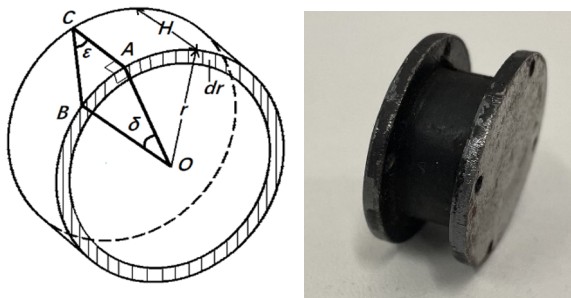

**Figure 3.** Rubber and the model figure of rubber torsion deformation.

Output torque $T$ of rubber is obtained by integrating micro torque $\tau$ on the radius $r$. If we integrate both sides of Equation (5) at the same time, the relationship between the torsion end face torque $T$ and the torsion angle $\delta$ can be obtained as follows:

$$\begin{cases} T = \int_0^R \tau dr \\ \frac{\eta}{E_2}\int_0^R \dot\tau dr + \int_0^R \tau dr = (\frac{E_1+E_2}{E_2}\eta\dot\delta + E_2\delta)\frac{2\pi}{H}\int_0^R r^3 dr \\ \frac{\eta}{E_2}\dot T + T = \frac{\pi R^4}{2H}(\frac{E_1+E_2}{E_2}\eta\dot\delta + E_2\delta) \end{cases} \tag{6}$$

where Zenner constitutive model parameters $E_1$, $E_2$ and $\eta$ are related to the material properties; it is relatively difficult to measure them directly. By the means of simplification, Equation (6) can be rewritten into Equation (7) to reduce the number of Zenner model parameters. These synthesis parameters are represented as relaxation coefficient $a$, damping coefficient $b$ and stiffness coefficient $c$ in Equation (7), and can be identified through system identification method:

$$a\dot T + T = b\dot\delta + c\delta \tag{7}$$

In order to facilitate parameter identification and ensure that the results are suitable to various conditions, the variation range of input signals should be large enough. The sinusoidal signal is often used as a test signal because of its continuous and regularly

varying amplitude. A sine signal, as shown in Equation (8), is taken as the excitation signal for system identification. When a sinusoidal excitation signal with frequency of 1 Hz and amplitude of 0.4 Nm is input to the rubber cylinder part, the deformation range can reach to $\pm 0.40$ rad and the deformation rate can reach to $\pm 2.41$ rad/s. This basically covers most of the expected operating conditions of our SVA.

$$x = 0.40 \sin(2\pi t) \tag{8}$$

The sampling period $T_s$ of output torque is set to 0.01 s, and Equation (7) can then be discretized as follows:

$$\begin{cases} T_k = \frac{a}{a+T_s} T_{k-1} + \frac{b+cT_s}{a+T_s} \delta_{k-1} - \frac{b}{a+T_s} \delta_{k-2} \\ (a_1 = \frac{a}{a+T_s}, b_1 = \frac{b+cT_s}{a+T_s}, b_2 = -\frac{b}{a+T_s}) \\ T_k = a_1 T_{k-1} + b_1 \delta_{k-1} + b_2 \delta_{k-2} \end{cases} \tag{9}$$

where $T$ and $\delta$ are output torque and rubber deformation. Supposing the parameter vector $\theta = [a_1, b_1, b_2]$ and the output vector $Y = [T_1, T_2, \dots, T_N]^T$, the least squares estimation of the parameter vector can be obtained as $\hat{\theta} = (\varphi^T \varphi)^{-1} \varphi^T Y$. We took the experimental data with sampling length $N = 1000$ to perform the parameter identification and got $[a_1, b_1, b_2] = [0.9873, 1.415, -1.34]$, accompanying with 0.028 Nm standard deviation error and 0.033 Nm residual error of estimation (the detailed results can be found in Section 3, Figure 10). Combining with Equation (9), the parameters $[a, b, c]$ of our rubber-based torque estimation model in Equation (7) can be obtained as $[0.7764, 1.055, 5.832]$. It is important to note that values of parameters depend on the rubber utilized.

### 2.4. Harmonic Drive Friction Compensation

Nonlinear friction of transmission part composed by harmonic drive is the main factor that affects the current-based torque estimation. As current-based torque estimation is the only observation in our SVA system, it is essential to compensate the friction caused by harmonic drive to improve the observation accuracy.

Harmonic drive friction is specifically generated between the wave generator and the flexible wheel, between the flexible wheel and the fixed rigid wheel, and between the shaft and the bearing. Establishing the friction model of the harmonic drive is the key to realize friction compensation. The most widely used one is the Stribeck exponential model, which describes the friction force as a function of speed, with a fitting accuracy of 90% for general friction force [31]. However, the transmission mode of the harmonic drive is complex and its friction characteristics are difficult to be described only as a function of speed.

The friction torque of harmonic drive has to be recorded if we want to build a detailed friction model. It is inconvenient to install a torque sensor to measure the friction torque directly in harmonic drive. As a result, an indirect measurement method based on motor current is adopted in our case. It can be indicated from Equation (2) that the motor current $I_q$ only responds to overcome the friction torque when the motor runs at a constant speed and with no load. Thus, the friction of harmonic drive can be obtained as follows:

$$T_f = NCI_q \tag{10}$$

In this way, $I_q$ was calculated through Clarke and Park transform by the three phase current collected on the motor drive board, and it can be used to indirectly estimate the friction torque under no-load steady working conditions.

In order to obtain the relationships between harmonic drive motion and friction, SVA is controlled to work at various velocities without load. The three-dimensional curves of friction torque, motor speed, and running time are depicted in Figure 4. Each curve represents the friction torque behavior at a specific speed. As the speed increases, the average amplitude of the friction torque shows an increasing trend, indicating that the friction includes a speed-related part. It can also be noted that the fluctuation trend of

friction changes is approximately sinusoidal at the same speed. In order to verify that this experimental phenomenon is mainly caused from the harmonic drive, rather than the cogging torque, we also measured the output torque of the motor without harmonic drive at the same speed shown in Figure 4. Experimental results shown in Figure 5 explained that the magnitude of the cogging torque is far less than that via using the harmonic drive. Considering that the angular position also changes periodically with time at a constant angular velocity, we assume that the dynamic friction torque is also related to angular position. Thus, the dynamic friction Tf of the harmonic drive is modeled as follows:

$$T_f = f(\dot{\varphi}) + A \sin \varphi \tag{11}$$

where the left term on the right side of Equation (11) represents the velocity-related part and the right term denotes the position-related part.

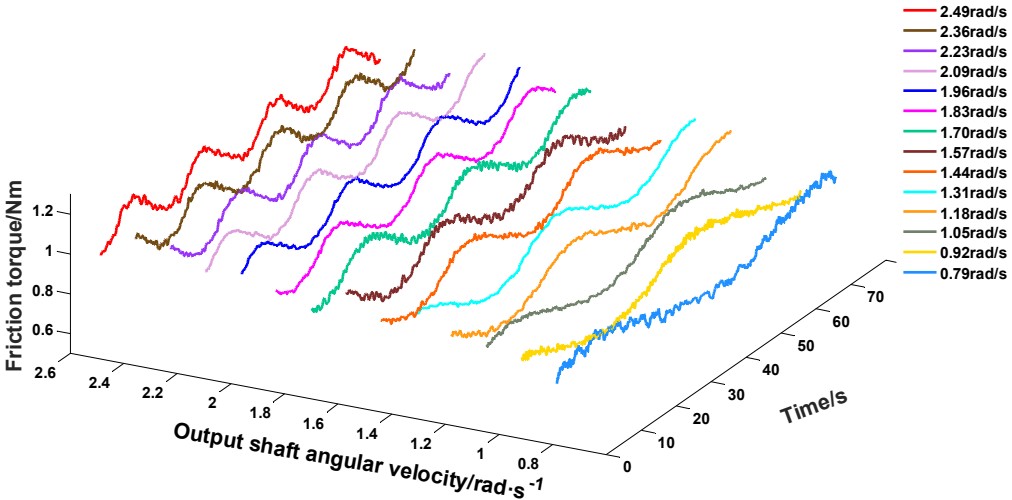

**Figure 4.** Torque-speed-time relationship of friction in harmonic drive.

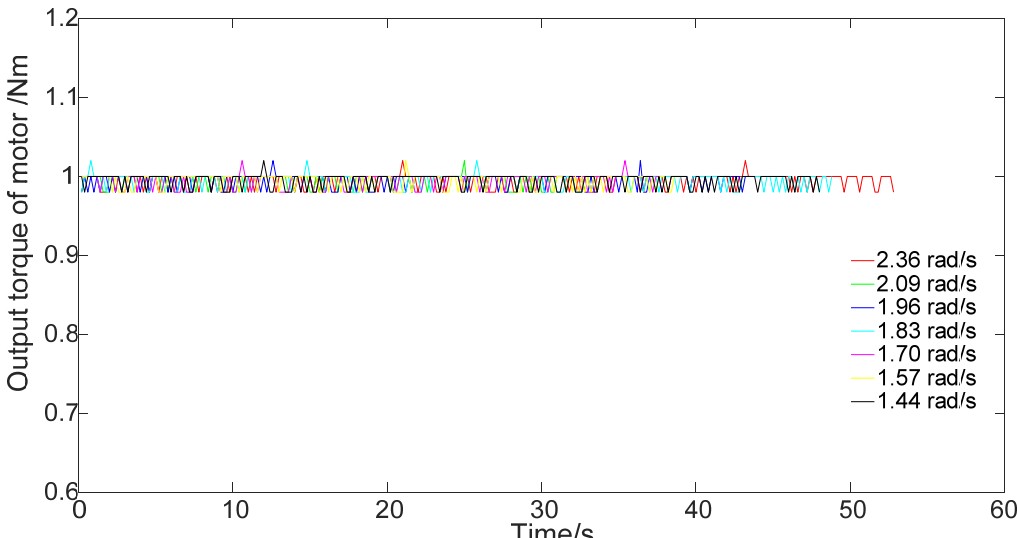

**Figure 5.** Output torque of motor without harmonic drive.

Currents and angular position data at speed of 1.31, 1.57, 1.83 and 2.09 rad/s are used to identify both terms in Equation (11). The velocity-related term is considered as an offset for Equation (11) and it can be calculated by averaging the friction value recorded in a sufficient long period. We input stepped angular velocity commands which ranged from 0.1~2.6 rad/s with step length of 0.1 rad/s and duration of 100 s. The curve of

velocity related friction torque was estimated (shown in Figure 6) to be compared with the Stribeck friction model (shown in Figure 7). It can be indicated from Figure 6 that the velocity-related friction fits Stribeck friction model well. Thus, Stribeck friction model, presented as follows, is adopted to model the velocity-related term.

$$f(\dot{\varphi}) = T_c + (T_{ms} + T_c)e^{-\left(\frac{\dot{\varphi}}{\dot{\varphi}_s}\right)^{\delta}} + \sigma\dot{\varphi} \tag{12}$$

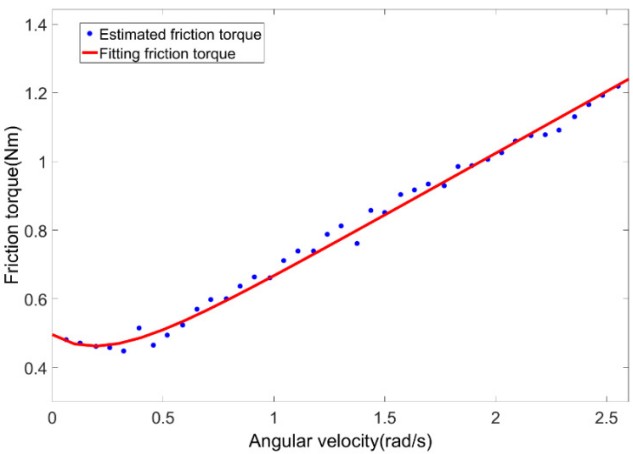

**Figure 6.** Friction torque with respect to harmonic angular velocity. In the low speed area (w < 0.3 rad/s), the friction torque will decrease with the increase of speed and accompany with serious nonlinearity. It is due to the fact that the maximum static friction torque is usually greater than the sliding friction torque. After reaching a certain speed (w > 0.3 rad/s), viscous friction plays a dominant role, and the friction torque increases approximately linearly with the angular velocity.

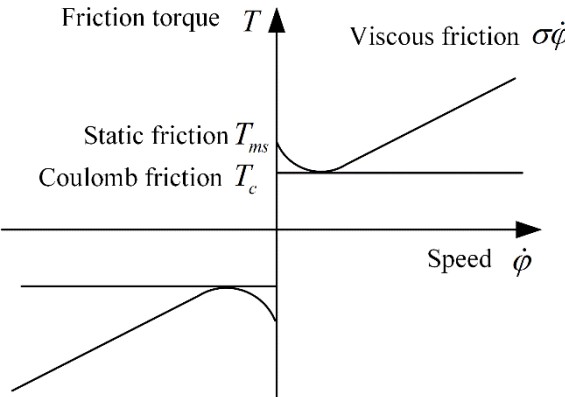

**Figure 7.** Stribeck friction model.

The parameters of the Stribeck friction model have clear physical meanings. $T_c$ is the Coulomb friction torque, $T_{ms}$ is the maximum static friction torque, $\sigma$ is the viscous friction factor, when $\dot{\varphi}$ is large enough, $\sigma$ is equivalent to the tangent slope, $\dot{\varphi}_s$ is the Stribeck velocity, $\dot{\varphi}_s$ is the abscissa of the intersection of the tangent line passing intercept $T_{ms}$ and the horizontal line passing through $T_c$, and $\delta$ is an empirical parameter.

In this paper, the value of $\delta$ is set to 1 according to the Tustin empirical model [32]. Although the model is a nonlinear function which is not convenient to system identification, most of the parameters can be determined directly through their physical meanings. $T_c$ can be obtained as 0.430 Nm from the intersection point of the horizontal line passing $\dot{\varphi}_s$ and the vertical axis in Figure 6. $T_{ms}$ which is 0.496 Nm in our case, can be obtained from the intercept in the first quadrant of the image in Figure 6. $\sigma$ (0.36 Nm·s/rad) can be obtained

from the slope of the image when $\dot{\varphi} > \dot{\varphi}_s$. The value of $\dot{\varphi}_s$ (0.25 rad/s) can be obtained from least squares estimation. When the angular velocity is reversed, $T_c$, $T_{ms}$, $\dot{\varphi}_s$ take the opposite number, and $\sigma$ remains unchanged.

The position-related friction torque value is obtained by subtracting the velocity-related term from Equation (11). The amplitude A of Equation (11) is obtained by fitting the results with sinusoidal function. In order to verify the relationship between amplitude A and angular velocity, each experiment is repeated three times under the conditions of four angular velocities mentioned above. The fitting results of the velocity-related term are shown in Table 1. The result of a single factor analysis of variance between A and $\dot{\varphi}$ is $p = 0.79 > 0.05$, which indicates that there is no significant correlation between the two parameters.

**Table 1.** Fitting parameters of dynamic friction model.

| $\dot{\varphi}$ (rad/s) | A (Nm) | | | $f(\dot{\varphi})$ |
|---|---|---|---|---|
| | 1 | 2 | 3 | |
| 1.31 | 0.101 | 0.100 | 0.102 | 0.776 |
| 1.57 | 0.103 | 0.098 | 0.098 | 0.870 |
| 1.83 | 0.098 | 0.101 | 0.101 | 0.963 |
| 2.09 | 0.097 | 0.103 | 0.097 | 1.056 |

Considering the direction of angular velocity and combining Equations (11) and (12), we can obtain the friction torque model of the SVA system as follows:

$$T_f = \begin{cases} 0.430 + 0.066\exp(-4\dot{\varphi}) + 0.36\dot{\varphi} + 0.1\sin\varphi & \dot{\varphi} \geq 0 \\ -0.430 - 0.066\exp(-4\dot{\varphi}) + 0.36\dot{\varphi} - 0.1\sin\varphi & \dot{\varphi} < 0 \end{cases} \tag{13}$$

Substituting Equation (13) into Equation (2), the output torque $T_l$ of SVA based on the motor current can be shown as follows:

$$T_l = \begin{cases} NCI_q - 0.430 - 0.066\exp(-4\dot{\varphi}) - 0.36\dot{\varphi} - 0.1\sin\varphi & \dot{\varphi} \geq 0 \\ NCI_q + 0.430 + 0.066\exp(-4\dot{\varphi}) - 0.36\dot{\varphi} + 0.1\sin\varphi & \dot{\varphi} < 0 \end{cases} \tag{14}$$

Since the current-based torque estimation is only reliable at steady motor working condition, a quantitative confidence function $\psi$ is defined in Equation (15) in order to judge the accuracy of the current estimation torque in a certain state. It represents the confidence in the accuracy of the current-based estimated torque state. The range of value of $\psi$ is [0, 1]. The closer to 0, the lower the accuracy and vice versa. It is calculated from the angular velocity $\dot{\varphi}$, angular acceleration $\ddot{\varphi}$, current rate of change $\dot{I}_q$, the constant Stribeck velocity $\dot{\varphi}_s$, and the scale factor $\gamma$. When the angular velocity $\dot{\varphi}$ is far away from the Stribeck velocity $\dot{\varphi}_s$, which characterizes the friction nonlinear region, its confidence is higher. When the angular acceleration $\ddot{\varphi}$ and the current change rate $\dot{I}_q$ are larger, angular velocity is more unstable and the product of the two is larger. As a result, the value of the calculation $\psi$ is lower. $\gamma$ is a scaling factor that facilitates unification of units and orders of magnitude. This state quantitative processing method is conducive to the design of subsequent data processing algorithms:

$$\psi = \max(0, \min(\frac{abs(\dot{\varphi}) - \dot{\varphi}_s}{\gamma \times \text{abs}(\ddot{\varphi}\dot{I}_q)}, 1)) \tag{15}$$

*2.5. Torque Estimation Based on Dual Extended Kalman Filter*

The rubber-based output torque estimation is influenced by two factors: nonlinearity and time variant of viscoelastic material properties. Nonlinearity affects torque estimation in a short time-scale while time-variant property affects the estimation results in a long time-scale. These two issues are attributed to the state estimation problem of the torque $T$ and

the parameter estimation problem of *a, b* and *c* in rubber model presented in Equation (7). To realize updating system state and parameters simultaneously, the state equations and observation equations of the state estimator EKF1 and the parameter estimator EKF2 need to be established, respectively. The traditional discrete Kalman filter a priori estimation equations and the measurement parameter update equations are given as follows:

$$
\begin{aligned}
&\text{Predict:} \\
&X_{k,k-1} = F_k X_{k-1} + B_k U_{k-1} + \omega_k \\
&P_{k,k-1} = F_k P_{k-1} F_k^T + Q_k \\
&\text{Update:} \\
&K_k = P_{k,k-1} H_k^T (H_k P_{k,k-1} H_k^T + R_k)^{-1} \\
&X_k = X_{k,k-1} + K_k (Z_k - H_k X_{k,k-1}) \\
&P_k = (I_{n \times n} - K_k H_k) P_{k,k-1}
\end{aligned}
\tag{16}
$$

where *X, U, F, B* and $\omega$ represent system state, system input, system matrix, input matrix and the external disturbance, respectively. *P, Q* and *R* are the covariance matrix of states, interfering noises and observed noise, respectively. *K, H* and *Z* respectively represent Kalman gain coefficient, observations matrix and observations equation.

For the state estimator EKF1, the state vector $X^t$ is set to be torque *T* directly in our case. The input vector *U* is set as $\left[ \dot{\delta}, \delta \right]$. Thus, state prediction equation for EKF1 can be obtained using a rubber model represented by combining Equation (7) with the first function in Equation (16) as follows:

$$
\begin{cases}
X_{k,k-1}^t = F_k X_{k-1}^t + B_k U_{k-1} + \omega_k^t \\
(F_k = \frac{a}{a+T_s}, B_k = [\frac{b+cT_s}{a+T_s}, -\frac{b}{a+T_s}])
\end{cases}
\tag{17}
$$

where the superscript *t* of the variable represents the Kalman equation with torque as the state variable. $T_s$ is the sampling time. *a, b* and *c* are the updated rubber parameters obtained from EKF2. $\omega$ and *k* represent the external disturbance and the present step.

The observation equation of EFK1 needs to be specially designed and processed because the feedback results are obtained from current-based torque estimation which is only reliable when motor works at steady state. The predicted state vector cannot be updated at the time when motor works at dynamic state or current-based torque estimation model is invalid. There are two ways to stop updating the state vector in the non-ideal state inspired by the fourth function in Equation (16). One is to set the motor current observations variance $R^t$ to be $\infty$ and the Kalman filter gain $K_k^t$ consequently becomes to 0, which means that there is no update in system state. Another way is to design the observation equation combining with a loss function [28] $\theta^t$ as follows:

$$
\begin{cases}
Z_k^t = H_k^t X_{k,k-1}^t + v^t \\
\theta^t = H_k^t X_{k,k-1}^t = \psi(T_{cur} - X_{k,k-1}^t)^2 \\
Z_k^t = \theta^t + v^t
\end{cases}
\tag{18}
$$

where $Z_k^t$, *v* and $\psi$ respectively represent real observations value, observed noise and quantitative confidence function defined in Equation (15). The loss function is selected in our case because changing $R^t$ parameter greatly during the operation process is not conducive to the stability of the updating algorithm.

As the loss function $\theta^t$ plays an important role in the observation equation, the predicted state vector will make the loss function to change toward the direction of 0 and finally converges to the real value if the actual observations value $Z_k^t$ is considered to be 0 at any time. When SVA is running in a non-steady state, $\psi$ is close to 0. No matter what the estimated current $T_{cur}$ is, $Z_k^t$ always equals to 0 and there is no updating for $X_k^t$. When the SVA is running in an ideal state, $\psi$ is close to 1 and $\theta^t$ can represent the deviation between

the estimated current $T_{cur}$ and the predicted $X_k$. If there is a deviation between $X_k^t$ and $T_{cur}$, $X_k^t$ will update towards $T_{cur}$, and the $\theta^t$ will become to 0.

Thus, the observation matrix of the state estimator EKF1 can be obtained:

$$H_k^t = \left.\frac{\partial[\psi(T_{cur} - X_k^t)^2]}{\partial X}\right|_{X=X_k^t} = -2\psi(T_{cur} - X_k^t) \tag{19}$$

For the parameter estimator EKF2, the rubber mechanics model parameters *a, b, c* are assumed to be constant in a short period of time. In EKF2, the state vector is $X^p = [a, b, c]^T$, and the state prediction equation is shown as follows:

$$X_k^p = X_{k-1}^p + \omega_k^p \tag{20}$$

where the superscript *p* of the variable represents the Kalman equation with the rubber parameter as the state variable.

The observations equation of EKF2 is similar with the form of EKF1 because the observations results are also obtained by current-based torque estimation. It is designed as a loss function $\theta^p$ of the current-based estimated torque $T_{cur}$ and the predicted torque predicted from the state vector $X_k$:

$$\theta^p = \psi[T_{cur} - (-\dot{T}a + \dot{\delta}b + \delta c)]^2 \tag{21}$$

where $\psi$ and T are, respectively, quantitative confidence function and torque obtained from EKF1. Equation (21) can be discretized as follows:

$$\theta^p = \psi\left\{T_{cur} - [-\frac{T_k - T_{k-1}}{T_s}, \frac{\delta_k - \delta_{k-1}}{T_s}, \delta_k]X_k^p\right\}^2 \tag{22}$$

In such case, the observations equation $Z_k^p$ of EKF2 can be acquired as follows:

$$\begin{cases} Z_k^p = H_k^p X_{k,k-1}^p + v^p \\ \theta^p = H_k^p X_{k,k-1}^p = \psi\left\{T_{cur} - [-\frac{T_k - T_{k-1}}{T_s}, \frac{\delta_k - \delta_{k-1}}{T_s}, \delta_k]X_k^p\right\}^2 \\ Z_k^p = \theta^p + v^p \end{cases} \tag{23}$$

where *T*, *k* and $\delta$ are the output torque of EFK1, the present step and the deformation variables, respectively. The design principle of the observations equation is the same as EKF1. It is considered that the actual value of the observations value is 0, and the state vector will be driven to the direction that makes the value of the loss function $\theta^p$ tend to 0. In such a way, the parameters a, b, and c will be iteratively updated until they converge to a true value. When these parameters change again due to temperature rising after long time use, the filter will continue to correct them to the changed true value. The square item in Equation (23) is represented by $e_k$ as follows:

$$e_k = T_{cur} - [-\frac{T_k - T_{k-1}}{T_s}, \frac{\delta_k - \delta_{k-1}}{T_s}, \delta_k]X_k^p \tag{24}$$

Then the observations matrix of EKF2 can be obtained by combining with Equations (23) and (24):

$$H_k^p = \left.\frac{\psi e_k^2}{\partial X}\right|_{X=X_k^p} = 2\psi e_k[-\frac{T_k - T_{k-1}}{T_s}, \frac{\delta_k - \delta_{k-1}}{T_s}, \delta_k] \tag{25}$$

The basic process of DEKF is as follows (as shown in Figure 8): Firstly, a priori parameter $X_{k,k-1}^p$ of $[a, b, c]_{k,k-1}^T$ is predicted by the parameter estimator EKF2 in one step.

These predicted parameters are used in the one-step prediction of the state estimator EKF1 to obtain a priori state $X_{k,k-1}^{t}$. Then, the update stage of EKF1 works to obtain the posterior state $X_{k}^{t}$ or the estimated torque $T_{k}^{t}$. In return, $X_{k}^{t}$ is utilized in the update phase of EKF2, and the posterior parameter $X_{k}^{p}$ of EKF2 can be obtained. In such a way, one iteration of the proposed DEKF calculation is executed and both the state $T$ and the parameter $[a, b, c]$ of SVA are estimated.

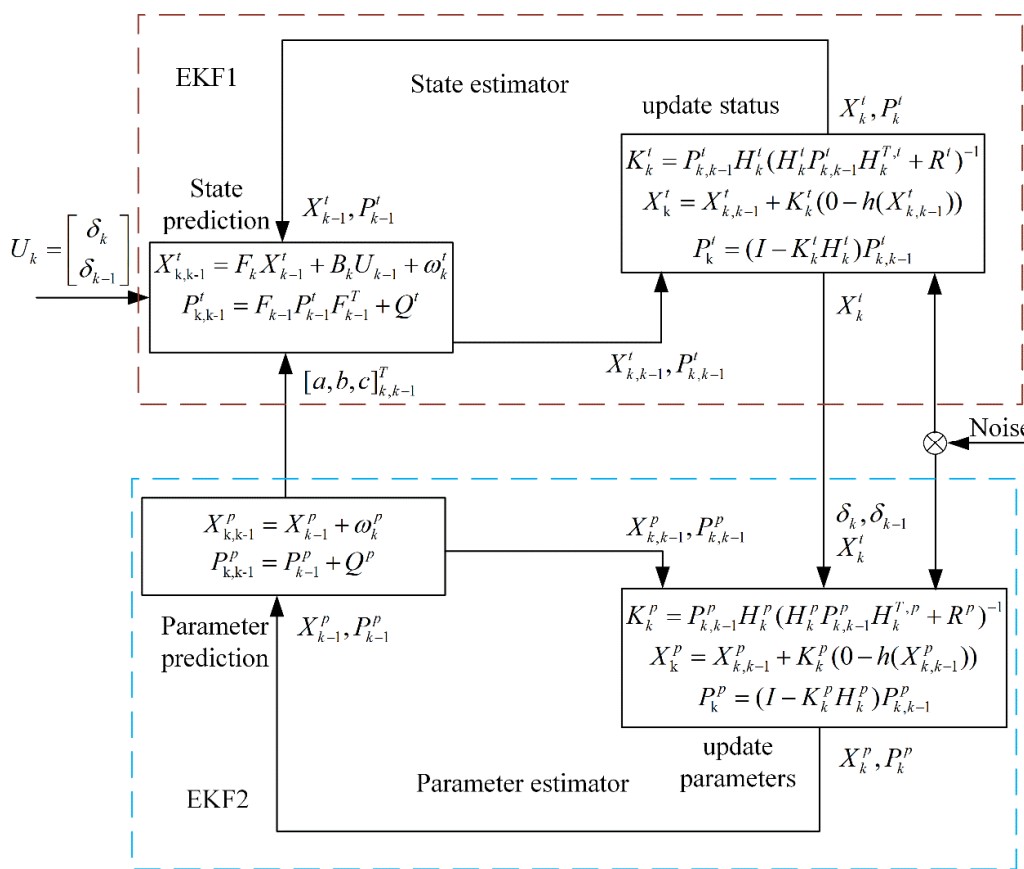

**Figure 8.** Workflow chart of proposed DEKF.

## 3. Experimental Setup and Results

### 3.1. Torque Measurement Experiment Using Rubber Alone

A torque estimation platform shown in Figure 9 has been built to verify the accuracy of the rubber torque estimation model in Equation (7). The test torque was generated by a BLDC motor (EC 45, Maxon, Sachseln, Switzerland) and applied to the rubber after amplifying 80 times by a harmonic drive (LCD-32, Leaderdrive, Suzhou, China). The deformation of the rubber can be obtained by calculating the angle differences of the two photoelectric encoders (0.04° resolution) mounted at both sides of the rubber part. The real torque is measured by a torque sensor (M2100A, Sunrise Instruments, Shanghai, China) with 10 Nm measurement range and 0.01 Nm resolution. The torque generated by the motor acted on the rubber to make the rubber move periodically within a range of ±0.37 rad. The torque produced by the motor was amplified by the harmonic reducer and transformed to rubber to make the deformation. The deformation of the rubber was obtained by the difference between the angles of the photoelectric encoder fixed at both ends of the rubber fixer. The actual output torque was measured by the torque sensor fixed on the right side of the device. We compare the actual torque measured by torque sensor and the torque calculated by Equation (7). The experimental results and torque error were shown in Figures 9 and 10, respectively. It can be seen from the experimental results that the estimated torque curve can accurately reflect the hysteresis characteristics of the rubber

element, and the estimated torque follows the actual torque well with calculated root mean square error of 0.03 Nm as shown in Figure 11.

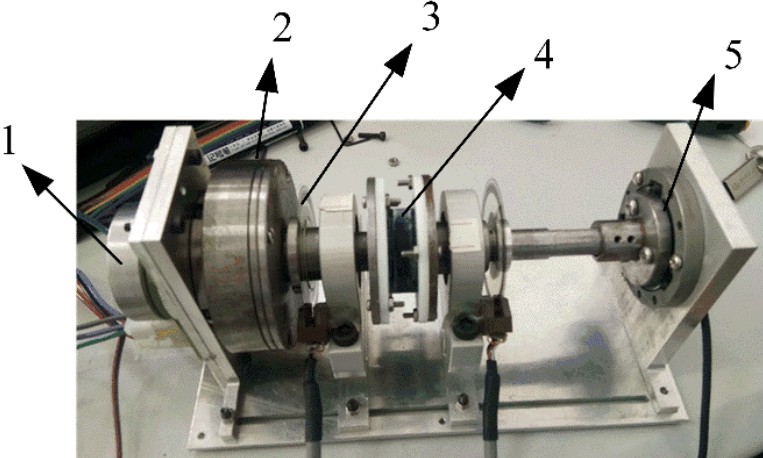

**Figure 9.** Rubber testing device: (1) motor, (2) harmonic drive, (3) photoelectric encoder, (4) rubber, (5) torque sensor. The left end of the rubber is connected to a motor, and the right end is fixed and connected to a torque sensor. Two encoders respectively record the angle changes at the left and right ends of the rubber. The difference between the two is used as the deformation of the rubber.

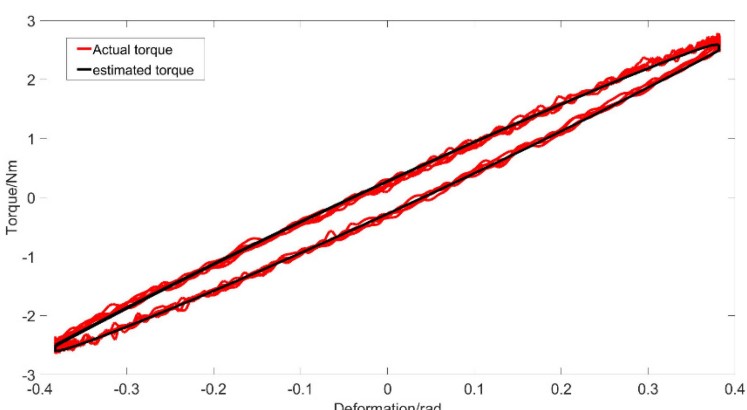

**Figure 10.** Rubber element torque deformation variable fitting curve. The estimated torque is in good agreement with the actual torque, which can accurately reflect the hysteresis characteristics of the rubber element.

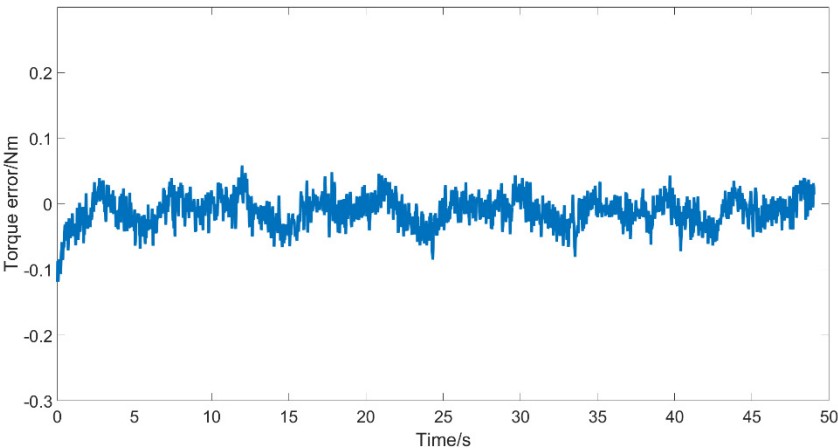

**Figure 11.** Torque-estimated error. The root mean square error of torque-estimated error is 0.03 Nm.

### 3.2. Torque Measurement Experiment Using Motor Current Alone

In order to verify the accuracy of the current-based torque estimation model as described by Equation (2), we applied a square wave reference command ($\pm$1.31 rad/s amplitude and 5 s width) to the SVA working under velocity control mode with PID feedback control. A random disturbance torque was exerted on the output shaft of SVA by hand to simulate the dynamic interaction with environment. Torque sensor fixed on the output shaft of SVA was used to measure the actual output torque. Comparing with the real torque value measured by the torque sensor, it can be seen from Figure 12 that the current-based estimation torque followed the real load torque well (2.23%/FS nonlinear error) in the angular velocity steady phase but the estimated torque had large oscillations (39.86%/FS nonlinear error) in angular velocity dynamic changing phase due to some factors such as remaining inertia and nonlinear friction when the angular acceleration changes drastically.

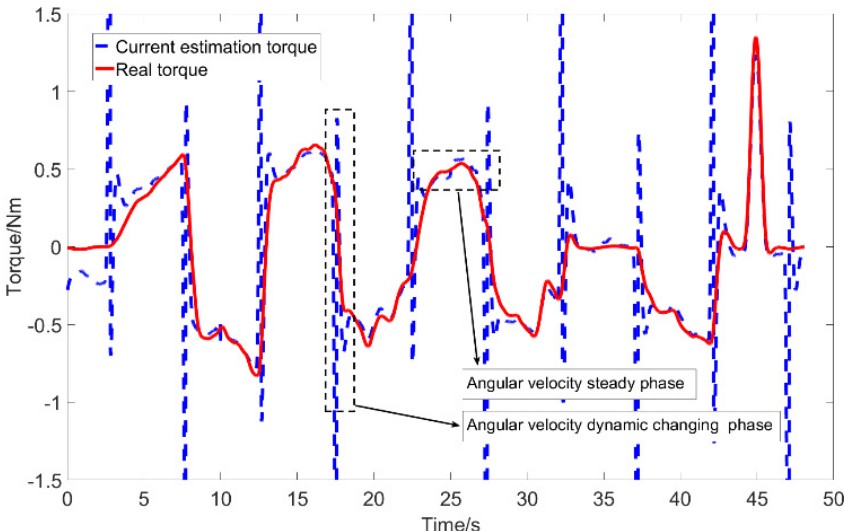

**Figure 12.** Comparison curve of current-based estimation torque and real torques. When the motor commutates or the angular acceleration is large, the output torque measured by the motor current has a large deviation from the real torque.

### 3.3. Torque Estimation Experiment Using DEFK

In order to verify the effectiveness of the proposed DEKF fusion algorithm, a short-period torque estimation experiment and a rubber model parameter changing simulation experiment were designed and performed on the SVA device. The control command and disturbance load to SVA were the same as in the current-based torque estimation method in Section 3.2. Initial parameter setting of the rubber model are divided into correct and incorrect group as shown in Table 2. The correct one is the parameter identification result of the rubber torque model in Section 2.3 and the incorrect one is an arbitrary value. Table 3 shows the current torque model parameters identified in Section 2.4. Table 4 shows the filter parameters in the DEKF model, including $P^t$, $Q^t$, $R^t$ of the state estimator EKF1 and $P^p$, $Q^p$, $R^p$ of the parameter estimator EKF2.

**Table 2.** Initial parameter setting of rubber model.

| Parameter Initial State | Relaxation Coefficient a (s) | Damping Coefficient b (Nm·s/rad) | Stiffness Coefficient c (Nm/rad) |
|---|---|---|---|
| Correct state | 0.7764 | 1.055 | 5.832 |
| Error state | 10 | 0.1 | 0.1 |

**Table 3.** Initial parameter setting of current model.

| Sine Amplitude A (Nm) | Coulomb Friction Torque $T_c$ (Nm) | Static Friction Torque $T_{ms}$ (Nm) | Viscous Friction Factor $\sigma$ (Nms/rad) | Stribeck Speed $\dot{\varphi}_s$ (rad/s) | Scale Factor $\gamma$ |
|---|---|---|---|---|---|
| 0.1 | 0.304 | 0.496 | 0.36 | 0.25 | 0.02 |

**Table 4.** DEKF initial parameter setting.

| $P^t$ | $Q^t$ | $R^t$ | $P^p$ | $Q^p$ | $R^p$ |
|---|---|---|---|---|---|
| 0.01 | 0.1 | 0.01 | 0.01 | 0.01 | 0.01 |

Firstly, the rubber model parameters were set to the previously identified result of [0.776, 1.055, 5.832] to verify the effect of the state estimator EKF1. These parameters are assumed to be correct in this experiment and the effect of EKF2 can be ignored since the experiment period is short. The experimental results of measured SVA output torque and estimated ones are depicted in Figure 13. In the whole process, the torque estimation error was small compared with using only current-based torque estimation method, and the root mean square error did not exceed 0.03 Nm. Although the motor speed changed frequently, the torque can be effectively estimated under these complex operating conditions. The experimental results indicate that the torque estimator EKF1 is effective.

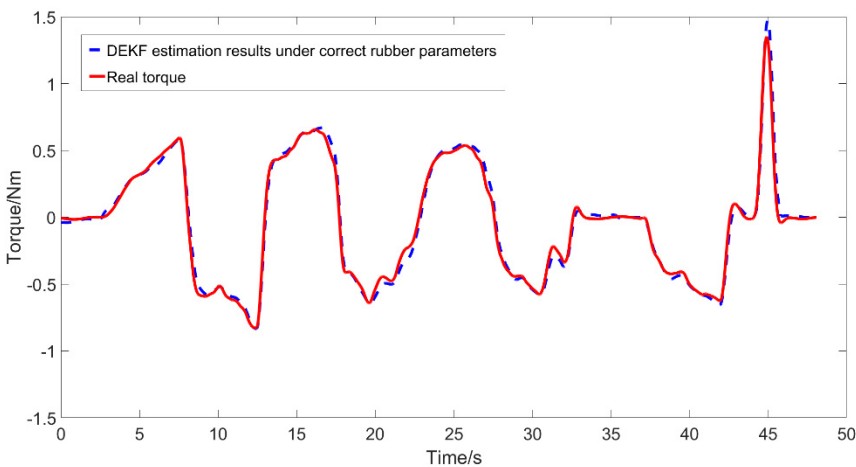

**Figure 13.** DEKF estimation results under correct rubber parameters.

Secondly, in order to verify the effect of the parameter estimator EKF2, the rubber model parameters [a, b, c] were set to the incorrect value of [10, 0.1, 0.1] which are deviated greatly from the identification result and used to simulate the time-variant property of rubber material under long periods of working conditions. The parameter estimation result using the DEKF is shown in Figure 14. It can be seen that the estimated torque deviated greatly from the real value due to the incorrect setting of the rubber element parameters from the real parameters at the beginning stage. The rubber element parameters were gradually estimated to be close to the correct value via the effect of EKF2. The output torque estimation error also decreases, eventually reduced to the same level compared with Figure 13.

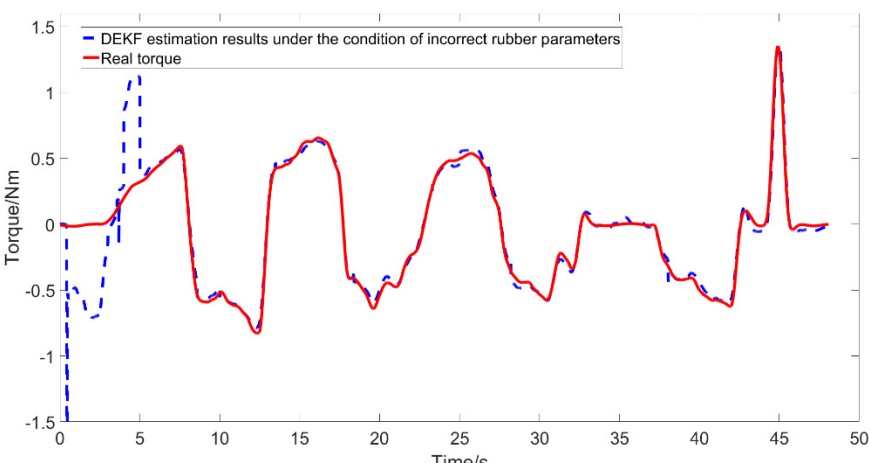

**Figure 14.** DEKF estimation results under the condition of incorrect rubber parameters.

The real-time output rubber model parameter's estimated value by the parameter estimator EKF2 during the processing was shown in Figure 15. Even if initial parameters' value of [10, 0.1, 0.1] with large deviation from the true one was given arbitrarily, EKF2 can calibrate it to be close to the true value of [0.7764, 1.055, 5.832]. These experimental results verified the effectiveness of the parameter estimator EKF2.

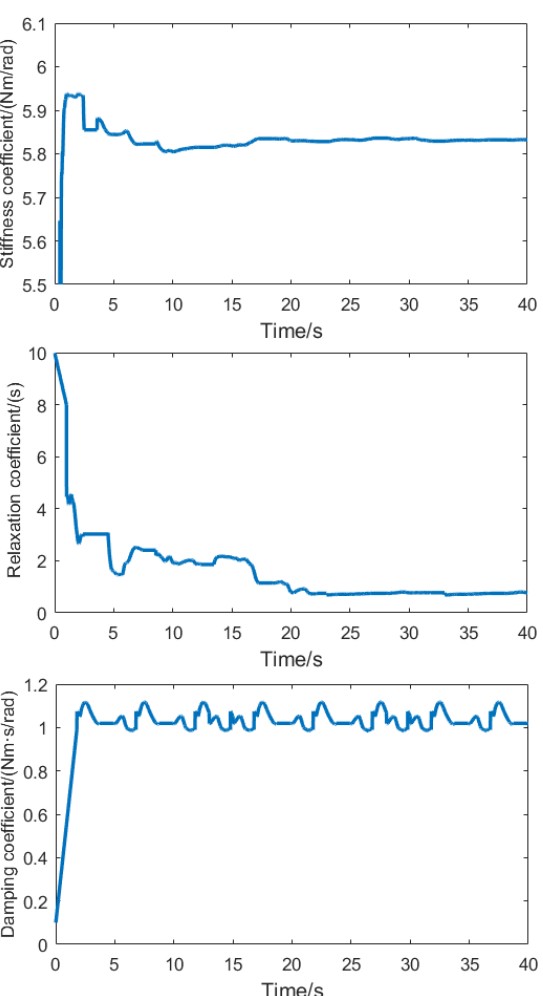

**Figure 15.** The real-time output rubber model-estimated parameter.

The above tests respectively verify the effectiveness of EKF1 for torque estimation and EKF2 for rubber parameter estimation, which indicates that the DEKF algorithm can overcome the shortcomings of current-based torque model which is unable to estimate the torque under complex operating conditions effectively.

### 3.4. Comparison of Three Torque Estimation Methods

Table 5 shows RMSE, coefficient of determination ($R^2$) and nonlinear error of the three torque estimation methods comparing with real torque. The calculation equation of nonlinear error is shown as follows:

$$e_n = (e_{max}/a_{fs}) * 100\% \tag{26}$$

where $e_n$, $e_{max}$, $a_{fs}$ represent the nonlinear error, maximum torque error between real torque and estimated torque, and the range of output torque, respectively. The overall effect of our proposed DEKF (0.027 RMSE and 1.48%/FS nonlinear error) was better than rubber torque estimation method (0.029 RMSE and 2.38%/FS nonlinear error) and current-based torque estimation method (0.178 RMSE and 18.6%/FS nonlinear error). Compared with DEKF and rubber-based torque estimation method, the RMSE of the current-based torque estimation were the largest due to the interference of harmonic drive nonlinear friction when motor worked at dynamic states such as commutation or suffering from impacts. Comparing with rubber-based torque estimation method, output torque estimated by DEKF method had smaller nonlinear error due to the fact that DEKF can constantly correct and update the rubber parameters close to the correct value.

**Table 5.** The estimation accuracy of the three torque estimation methods.

| Torque Estimation Methods | RMSE (Nm) | $R^2$ | Nonlinear Error (%/FS) |
|---|---|---|---|
| Rubber estimation | 0.029 | 0.9911 | 2.38 |
| Current estimation | 0.178 | 0.2759 | 18.6 |
| DEKF estimation | 0.027 | 0.9935 | 1.48 |

### 4. Conclusions

In order to improve the accuracy of SVA torque estimation, this paper proposed a dual extended Kalman filter-(DEKF) based torque estimation method, and respectively modeled the rubber-based torque estimation model and the dynamic friction of harmonic drive. The experimental results showed that the estimated torque by DEFK torque estimation method can effectively suppress the torque error of SVA system for non-constant acceleration conditions. By comparing the results of only using rubber-based torque estimation method and only using current-based torque estimation method with DEKF method, it was found that DEKF torque estimation method can effectively reduce the error of estimated torque, which proved the feasibility of this method. In the near future, we will intend to realize the impedance control of SVA and take it as the robot joint. In such a case, the torques measured by DEFK could be used as relatively accurate robot joint feedback torques.

**Author Contributions:** Writing—original draft and Data curation, H.W.; Conceptualization and Investigation, K.X.; Formal analysis, H.C.; Software, B.T.; Writing—review and editing, M.P. All authors have read and agreed to the published version of the manuscript.

**Funding:** This work was supported in part by the National Natural Science Foundation of China under grant numbers 61603284 and 61903286.

**Institutional Review Board Statement:** Not applicable.

**Informed Consent Statement:** Not applicable.

**Data Availability Statement:** Not applicable.

**Acknowledgments:** This work is financially supported by National Natural Science Foundation of China under grant numbers 61603284 and 61903286. The authors declare that we have no conflicts of interest regarding this work.

**Conflicts of Interest:** The funders had no role in the design of the study; in the collection, analyses, or interpretation of data; in the writing of the manuscript, or in the decision to publish the results.

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
