# Peer review of "Improvement of Torque Estimation for Series Viscoelastic Actuator Based on Dual Extended Kalman Filter"

_actuators, doi:10.3390/act10100258_

Round 1
Reviewer 1 Report
In this paper, a dual extended Kalman filter (DEFK) is proposed to estimate the force of Series viscoelastic actuators (SVA). By using DEFK method, the authors can estimate the force effectively. The presented methods are experimentally validated, and could be a good example for other researchers to deal with the similar problems. Generally, the idea of DEKF is interesting and the techniques developed are solid. However, the work in this paper should be further improved before consideration of publication.
Comment 1: The authors use the DEFK method to estimate the force, but EKF is a traditional method in the area of state estimation. Therefore, the contributions and novelties of this paper should be emphasized.
Comment 2: It can be seen that there are two EKF in the proposed method, the authors are suggested to make a clear description about the difference of two EKF in the application.
Comment 3: In page 4, authors state that “the q-axis torque current Iq decoupled from the field-oriented control algorithm”, how can be a current decoupled from an algorithm. Authors should give an explanation to this point.
Comment 4: In experiment part, the authors have given the experiments setup, but it is lack of general description for the experiments setup. The authors should add a full description to explain the experiment.
Comment 5: In the part of conclusion, the authors are suggested to give more description about this method and the future work.
Comment 6: There are some spelling mistakes in this manuscript, such as in page 3, there are a dot in front of the first sentence of section 2.1.
Reviewer 2 Report
This manuscript describes a method to estimate actuation torque from measurements of motor current and rubber elongation using a two-stage extended Kalman Filter for a Series viscoelastic actuators (SVAs). The main contribution is the design of the Kalman Filter and the experimental validation of the approach. The topics of SVAs and torque estimation are of great interest to the community and the authors present an interesting discussion on the modeling details inherent in their approach. The paper is well written and conveys most of the ideas logically. There are writing errors that have to be improved (see some examples below).
## Minor details
-- In the first sentence of the abstract: "Compared to series elastic actuators (SEA), Series viscoelastic actuators (SVA) is more 8 efficient with respect to higher force control bandwidth and suppression of high frequency oscillations induced by environment." It is unclear if "more efficient respect to higher force control bandwidth" refers to energy efficiency or any other kind. Also, it is misleading to state that SEAs may be worse than SVA with respect to higher force control bandwidth. This statement depends on the stiffness of the spring (which could be arbitrarily rigid) and the motion of the load. Consider rewriting this initial sentence.
-- In the introduction, "Motor current could also be used to estimate the output torque of SVA if the actuator works at steady conditions..." Can you elaborate on what you mean by steady conditions? Estimation of motor torque for a non-constant acceleration of the motor using motor current is accurate as well, e.g., [2]. Seems like the requirement of steady-state refers to the estimation of motor torque from motor currents when there is a harmonic drive. If this is the case, consider rewriting the sentence to make explicit that this statement applies to the case of harmonic drives.
-- The methods section refers to "Rubber material" please specify the vendor and reference of the rubber material. If you cured the rubber, please provide details of the curing process (e.g., percentages of materials, time and temperature in the oven).
-- In Section 2.3, "We took the experimental data with sampling length N=1000 to perform the parameter identification and got [a1,b1,b2]=[0.9873, 1.415, -1.34] after calculating." Please report the residual and the standard deviation of the estimation error to have insight into the uncertainty of the parameter estimation.
-- In Section 2.4 the authors use the Iq current to measure the friction torque from the harmonic drive. However, the authors do not show any experiments validating the accuracy of the estimation of motor torque from motor current. In addition, the authors mention: "In this way, Iq which is recorded by sampling resistor in the motor driven circuit board". This is misleading as the Iq current is never produced by the motor driver, it is a computed value that depends on the motor magnetic orientation and its relative position to the A,B,C motor windings.
-- In Section 2.4 "Because the angular position also changes periodically with time at a constant angular velocity, we assume that the dynamic friction torque is also related to angular position." Although this may be true, the authors should perform a test without a harmonic drive to check if the dependency of torque to the angular position is not due to cogging torque [3].
-- In the opinion of this reviewer, it is useful to know that the DEFK improved the torque estimation even for non-constant acceleration conditions. Please consider adding a short comment on this result within the conclusion section.
-- In the presented results, most of the inaccuracies for the estimation of motor torque from motor current are due to the harmonic drive. The authors should make this point explicit to avoid conveying the idea that torque estimates from motor currents are inaccurate.
## English writing
-- In the abstract, "SVA has recently". The acronym SVA refers to a plural noun.
-- In the abstract, "linear viscoelastic model 17 to model the the viscoelastic element of the rubber" eliminate double "the".
-- The first sentence of Section 2 starts with ".".
-- In Section 2, "The motor and harmonic drive is..."
-- Close to line 162, "In 162 such situation, equation (1) is degraded to eqaution (2)"
-- Line 177 "Figure3"
-- Line 203 "equation(7)"
-- Line 444, "An torque sensor was fixed on the output shaft of SVA"
## References
[1] D. Rollinson, S. Ford, B. Brown, and H. Choset, “Design and Modeling of a Series Elastic Element for Snake Robots,” in Volume 1: Aerial Vehicles; Aerospace Control; Alternative Energy; Automotive Control Systems; Battery Systems; Beams and Flexible Structures; Biologically-Inspired Control and its Applications; Bio-Medical and Bio-Mechanical Systems; Biomedical Robots and, 2013, vol. 1, pp. 2–6.
[2] T. Elery, S. Rezazadeh, C. Nesler, and R. D. Gregg, “Design and Validation of a Powered Knee–Ankle Prosthesis With High-Torque, Low-Impedance Actuators,” IEEE Trans. Robot., pp. 1–20, 2020.
[3] L. Xin, Z. Dong, Z. Pei and W. Rui, "The current feedforward compensation method for the cogging torque of permanent magnet synchronous motors," 2019 IEEE 9th Annual International Conference on CYBER Technology in Automation, Control, and Intelligent Systems (CYBER), 2019, pp. 1091-1095, doi: 10.1109/CYBER46603.2019.9066631.
Round 2
Reviewer 1 Report
This manuscript has been revised according to reviewer's comment, I think it can meet the publication requirement of this journal. Here it follows a list of aspects which, in my opinion, need to be addressed: Some relevant recent works of the state of the art have been overlooked, especially for SEA, VSA design. [1] Sun, J., Guo, Z., Sun, D., He, S., & Xiao, X. (2018). Design, modeling and control of a novel compact, energy-efficient, and rotational serial variable stiffness actuator (SVSA-II). Mechanism and Machine Theory, 130, 123-136. [2] Sun, J., Guo, Z., Zhang, Y., Xiao, X., & Tan, J. (2018). A novel design of serial variable stiffness actuator based on an archimedean spiral relocation mechanism. IEEE/ASME Transactions on Mechatronics, 23(5), 2121-2131.Author Response
Please see the attachment
